# The Effect of Eight-Week Functional Core Training on Core Stability in Young Rhythmic Gymnasts: A Randomized Clinical Trial

**DOI:** 10.3390/ijerph19063509

**Published:** 2022-03-16

**Authors:** Cristina Cabrejas, Mónica Solana-Tramunt, Jose Morales, Josep Campos-Rius, Alberto Ortegón, Ainhoa Nieto-Guisado, Eduardo Carballeira

**Affiliations:** 1Department of Sports Sciences, Ramon Llull University, FPCEE Blanquerna, 08022 Barcelona, Spain; cristinacm19@blanquerna.url.edu (C.C.); josema@blanquerna.url.edu (J.M.); josepcr@blanquerna.url.edu (J.C.-R.); albertoop@blanquerna.url.edu (A.O.); ainhoang@blanquerna.url.edu (A.N.-G.); 2Royal Spanish Swimming Federation, 28007 Madrid, Spain; 3NSCA, 28036 Madrid, Spain; 4Department of Physical Education and Sport, Faculty of Sport Science and Physical Education, University of A Coruna, 15179 A Coruña, Spain; eduardo.carballeira@udc.es

**Keywords:** rhythmic gymnastics performance, core motor control, lumbopelvic motor control, lumbopelvic-training, pelvic tilt test, bent knee fall out, active straight leg raise

## Abstract

It is suggested that core stability (CS) might improve rhythmic gymnasts’ performance. Nevertheless, the effect of core stability training (CST) in CS performance is not clear. Purpose: Evaluating the effect of an eight-week functional CST on young rhythmics gymnasts’ CS performance. **Method:** A sample of 45 young female rhythmic gymnasts from a competitive team (age = 10.5 ± 1.8 years, height = 144.1 ± 10.6 cm, weight 38.2 ± 8.9 kg, peak height velocity (PHV) = 12.2 ± 0.6 years) participated in the study. The participants were randomly allocated into the control group (CG) and experimental group (EG) and completed pre-tests and post-tests of specific CS tests using a pressure biofeedback unit (PBU). The CS was assessed by the bent knee fall out (BKFO), the active straight leg raise (ASLR) tests and the pelvic tilt test, all performed on the right and left sides. The EG (n = 23) performed an eight-week functional CST program based on rhythmic gymnastics (RG) technical requirements added to the traditional RG training sessions. Meanwhile, the CG (n = 22) received the traditional RG training sessions. **Results:** Mixed model analysis showed non-significant interaction effects; however, the ANOVA omnibus test showed a time effect (*p* < 0.05) in right BKFO (F_1,42_ = 4.60; *p* = 0.038) and both pelvic tilt tests (right F_1,42_ = 22.01, *p* < 0.001; left F_1,42_ = 19.13, *p* < 0.001). There were non-significant interaction effects. The fixed effects estimated parameters for right BKFO showed that both groups had less pressure variation after intervention compared with pre-intervention (β = −1.85 mmHg, 95%CI = [−3.54 to −0.16], t42 = −2.14, *p* = 0.038). Furthermore, the left pelvic tilt (β = 37.0 s, 95%CI = [20.4 to 53.6], t_42_ = 4.37, *p* < 0.001) improved 8.9 s more than the right pelvic tilt (β = 28.1 s, 95%CI = [16.3 to 39.8], t_42_ = 4.69, *p* < 0.001) considering both groups together. **Conclusions:** Adding a functional CST to regular training showed a trend in improving the performance of CS-related variables, which could help improve RG-specific performance. Coaches working with rhythmic gymnasts should consider adding a functional CST to regular training to improve CS performance leading to increased specific RG performance.

## 1. Introduction

Rhythmic gymnastics (RG) is a sport that requires early selection of athletes and intensive training in childhood and adolescence [1,2]. Important performance predictors for RG novices are strength and, in general, physical fitness [3,4]. Rhythmic gymnasts also require postural control to maintain the balance elements of the body difficulty section of the code of points [5]. RG balance exercises require holding a specific body position with a minimal surface; hence RG requires high demands towards postural control [6].

Core stability (CS) is the ability to control the position and movement of the trunk over the pelvis [7]; therefore, it has an important function over postural control. The core is a muscular corset that works as a unit to stabilize the body and especially the spine [8]. CS is the product of motor control and muscle capacity to produce strength of the lumbopelvic-hip complex [9,10]. To have a good CS is indispensable to efficiently transmit force by the lumbopelvic-hip complex [9], providing trunk strength, static and dynamic balance [7,11]. An enhanced CS help to transmit forces between upper and lower limbs efficiently and increase performance in situations when postural control is highly demanded [7,12].

Core stability training (CST) in high-level activities should be enabling performance while keeping the spine stabilized [13]. CST programs used for therapeutic purposes may not be helpful for athletic performance since they are mainly focused on preparing the patient to perform everyday tasks without pain, maintaining an aligned posture without challenging forces [14,15]. On the other hand, in sports performance, the core should absorb the large impacts, for example in the reception of jumps, and in transmitting the forces efficiently to the extremities [7,12,16]. In athletic environment, CS involves dynamically controlling and transferring large forces from the upper and lower extremities through the core to maximize performance and promote efficient biomechanics [17]. It is accepted that CS training programs have to respect the functional characteristics of the sports to be transferable [7,18]. Thus, CST exercises are functional for a specific sport when these exercises lead to an efficient and specialized motor unit recruitment to achieve the proper coordination of the segments involved in the kinetic chain of sport-specific skills [18]. Furthermore, CS exercises performed in conjunction with plyometric exercises, are recommended to improve sports performance [19]. Several studies evaluated the effect of CST on sports performance. Most of them are purposes from a length of 4 to 12 weeks; 8-week programs are the average length [20,21,22,23,24,25,26,27]. These findings provide a basis for further research to train and evaluate the specific role of CS in performance. Even though a few studies found an association between CS and athletic performance [22,23,24,28], nevertheless, it is accepted that there is still a lack of more functional CST programs and assessments with sufficiently sensitive measurement protocols [29].

To the best of our knowledge, CS exercises employed recently in CST for rhythmic gymnastics do not provide specific stimulus to technical movements [18,25,26]. Therefore, this study aimed to evaluate the effect of an eight-week of functional CST on the CS of young rhythmic gymnasts. This study hypothesized that an eight-week intervention, consisting of functional CST integrating core stability and strength actions while performing different specific RG movements, jumps and postures, could be a valuable method for improving the CS in young rhythmic gymnasts.

## 2. Materials and Methods

### 2.1. Design

Randomized parallel clinical trial.

The study was conducted according to the CONSORT standards [30].

### 2.2. Participants

The GPOWER v3.1 software (Bonn FRG, University of Bonn, Department of Psychology, Düsseldorf, Germany) was used to calculate a priori the sample size necessary to obtain a Power (1 − ß) > 0.9, effect size = 0.4 and α = 0.05, the result was from a required total sample of 36 subjects. Finally, the sample was established at 45 initial participants in anticipation of possible sample loss.

Forty-five young female rhythmic gymnasts participated in the current study. All gymnasts competed in RG regional federated and school competitions and trained in an RG national level competitive team (Figure 1). Exclusion criteria were: (1) gymnasts who trained less than three sessions per week, (2) had less than one year of experience in the competition or, (3) having any pain or injury that could disturb the testing or training development.

Participants were randomly assigned following simple randomization procedures (computerized random numbers) and were allocated in two groups: the experimental group (EG, n = 23) and the control group (CG, n = 22), (Figure 1).

Participants could not intake any drink or medicine that could disturb the nervous system. Descriptive characteristics of the participants are shown in Table 1. No significant differences were found in the demographic and anthropometric characteristics between groups before the testing session.

According to the latest version of the WMA Declaration of Helsinki, the subjects were fully informed and provided their written informed consent before participating in the study regarding the experimental procedures and the potential risks. The ethics committee of the Ramon Llull University of Barcelona approved the conduct of this study (approval number: CER- FPCEE Blanquerna, 1819007D). All participants’ respective parents or legal guardians completed the informed consent document prior to the study.

### 2.3. Procedures

The EG underwent the 8-week functional program with 3 sessions per week. The CG followed the traditional training across the same sessions per week.

The EG completed the integrated functional CST at the same time of the day as their scheduled training from 19 March 2020 until 15 May 2020. The duration of the specific training was approximately 30 min for eight weeks with a total of 24 sessions. One RG qualified professional observed the EG gymnasts while performing the exercises to ensure the training was performed correctly. All participants were tested one week before and one week after the 8-week intervention. Before the testing session, the subjects’ order and the applied tests’ order were randomly determined using a true random number generator to control bias. The CS tests were carried out in one day in the pre-test and post-test by a sport-specialized physiotherapist, unaware of the subject’s allocation group. The gymnasts performed a 15 min warm-up before the tests which consisted of cardiovascular activation and stretching exercises.

Anthropometric measures were gathered on a second day in the pre-test and post-test by the same physiotherapist. These measures helped us estimate gymnasts’ biological maturity throughout predicting the peak height velocity (PHV) age, using the non-invasive technique proposed by Mirwald et al. [31].

Integrated functional CST protocol.

The present study was registered in clinicaltrials.gov (accessed on 29 January 2022) with the number NCT04663633. CST program involved exercises that challenge gymnasts’ balance and postural control and comprised stretch-shortening cycle (SSC) explosive strength exercises, executed with specific RG elements and postures, with higher loads than the traditional training (Table 2).

All participants were asked to keep their lumbopelvic area straight and stable while performing different specific actions as jumps and specific balance elements. They were encouraged to hollow the navel while exhaling during each breath and each repetition, as repetitions were count by breathing cycles.

Participants rated the intensity of the sessions throughout a rate of perceived exertion (RPE) session scale (sRPE), a valid method of quantifying exercise training during a wide variety of types of exercise [32]. A CST trial was applied to check the gymnasts’ perceived exercise intensity. The load to achieve a prescribed number of repetitions was adjusted to 7–8 values in the RPE scale (i.e., hard). Thirty minutes after every CST session, all gymnasts (EG and CG) scored on the sRPE scale [32]. To obtain the sRPE the score was multiplied by the minutes of the session [33]. This outcome was used to modulate the training periodization plan. When values were lower than 7–8 sRPE a set was added in the exercises that were technically well executed. Similarly, the maintenance time of correctly executed isometric exercises was doubled in sRPE bellow 7–8.

The functional integrated CST contained three blocks: the first block was composed of a specific skill circuit with unstable surfaces, cones and hurdles; the second block included plyometric exercises in specific RG balances and jumps combined with CS actions; and the third block included a mixed exercises of specific balance postures in a core demanding lying position and jumps. We selected three different balances and leaps that are very common in RG (novice to intermediate competitive level) to convert the core and the integrated CST exercises into sport-specific functional training. The balances selected were the passé balance, the side balance with help, and the arabesque. The three leaps were the scissors, the stag from assemble, and the split leap. The mentioned RG elements were selected due to their lower limb position planes variety and different techniques since it is advised that a range of exercises be performed to challenge the core musculature in all three planes and ranges of movement to fully develop the CS [34]. Moreover, all exercises were executed equitably for the right and left sides (exercises of each block are shown in Appendix A). We included unstable surfaces and softballs to perform core and balance exercises (Figure 2). The gymnasts performed three exercises of the circuit on an unstable surface, aiming to stimulate anticipatory adjustments of the stabilizing muscles while trying to minimize postural destabilization [35]. The investigators (an expert CS physiotherapist and a professional RG coach) developed the training protocol and exercises, and professional RG coaches conducted it.

During the eight-week CST intervention, both the EG and the CG performed a warm-up together, consisting of general activation and stretching (30 min approximately). Afterwards, the EG went through the CST and the CG realized the usual conventional RG warm-up, which consisted of a specific warm-up, combining flexibility, strength, classical non-functional core exercises (as crunches, arm/leg raises and planks), and RG body technique on the floor. Once this part of the session was finished, both groups underwent the regular training planned for every day.

### 2.4. Measurements

#### 2.4.1. Instruments and Testing Procedures

A Pressure Biofeedback Unit (PBU) was used to assess lumbopelvic motor control (LPMC) before and after the intervention [36,37]. The PBU (Stabilizer^®®^, Chattanooga Group, Inc., Hixson, TN, USA) is a reliable, non-invasive, non-painful device consisting of a combined gauge/inflation bulb connected to a pressure cell. This device registers changes in pressure in an air-filled pressure cell during different lower limb movements. The gauge contains 16.7 × 24 cm of non-elastic material. The pressure cell measures from 0 to 200 mmHg, with a precision of 2 mmHg. Changes in body position modify the pressure, which is registered by the sphygmomanometer [38]. The PBU is an unexpensive device that can help practitioners to have an objective data about the ability of the RG to keep their lumbopelvic area stable while they are moving their lower limbs, as they do on their sport.

A manual chronometer (Namaste© model 898, Spain) was used to quantify the 10 s duration of the ASLR test and the total time performed by the participants in the pelvic tilt test.

Two PBU tests were performed in order to assess LPMC:

#### 2.4.2. Active Straight Leg Raise (ASLR) Test

The ASLR was performed in a supine position according to Solana-Tramunt et al. [39]. The inflatable pad was placed horizontally under the lumbar spine of the participant, with the lower edge at the level of the posterior superior iliac spines and was inflated to 40 mmHg. The subject was indicated to lift one extended lower limb 20 cm and hold it for 10 s. The positive or negative absolute mmHg deviation was registered for analysis. ASLR tests were performed on the right and left sides.

#### 2.4.3. Bent Knee Fall Out (BKFO) Test

For the BKFO, the subjects were positioned equally in a supine position. The participants flexed both knees by 120° and were asked to slowly bend their hip to approximately 45° of abduction and external rotation while keeping their other limb in a neutral position and then return to the starting position, repeating the movement three times. Two joined PBUs were placed under the center of the back in L3 level and connected along the spine to avoid differences in the lumbar tactile cue, although only the data of the PBU from the moving limb were considered [39]. The positive or negative absolute mmHg deviation was registered for analysis. BKFO tests were performed on the right and left sides.

Additionally, the pelvic tilt test was used to assess the endurance of the muscles responsible to keep the lumbopelvic area stable in neutral position [40]. We add the Pelvic Tilt test to check the endurance of the muscles who are responsible to maintaining the lumbopelvic area aligned.

#### 2.4.4. Pelvic Tilt Test

The gymnasts were positioned supine while lying on a thin mat. The participants flexed one knee by 120° while the other lower limb was extended. The gymnasts were asked to keep the limb straight aligning the shoulders, hip, knee, and ankle while keeping both knees together. The chronometer was activated once the subject fixed the position, and the leveler (Measure app from apple) was placed over the stomach to control that the pelvis inclination did not exceed the 10° difference from the starting position. The participants were instructed to hold up until their endurance limit. The chronometer was paused when the gymnasts’ pelvis contacted the ground or there was more than 10° difference in the pelvis position. Pelvic tilt tests were performed on the right and left sides. The time in seconds was recorded for analysis.

#### 2.4.5. Peak Height Velocity (PHV) Age

To estimate biological maturation and to distinguish whether changes in physical performance are due to maturation or exposure to regular exercise training, the PHV was calculated [41]. PHV age was predicted through a multiple regression equation using the anthropometric measures of standing height, sitting height, leg length, and weight. The equation calculates the time interval in years between the predicted age at PHV and the individual’s current age. The values can be negative if the age of PHV has not been reached yet, positive if the age of PHV has passed, or zero (0) if the current age is the exact age of PHV [31].

### 2.5. Statistical Analyses

Participants’ descriptive data are presented as mean ± standard deviation (SD). Descriptive data from the inferential analysis are the estimated marginal means with a 95% confidence interval (CI; lower limit to upper limit). Normality was assessed through standard distribution measures, visual inspection of Q–Q plots and box plots, and the Shapiro–Wilk test. Since our dependent variables were no-normally distributed, dependent variables outcomes were not expected to have a constant variance across time points, therefore we employed linear mixed models for longitudinal data to analyze changes within and between groups in dependent variables [42].

We employed the module GAMLj [43], which uses the R formulation of random effects as implemented by the function lme4, an R package, in Jamovi software (The jamovi project, v1.6, 2021). GAMLj estimates variance components with restricted (residual) maximum likelihood (REML), producing unbiased estimates of variance and covariance parameters, unlike earlier maximum likelihood estimation. The inter-subject factor group (EG and CG), the intra-subject factor time (pre- and post-intervention), and the interaction (group × time) were set as fixed effects. The participants’ intercepts were set as a random effect. Within-subject and between-subject changes were first evaluated by ANOVA F omnibus test employing the Satterthwaite approximation of degrees of freedom and secondly estimating the coefficients with their 95% CIs for the fixed effects in the mixed model. Furthermore, the variance of the random coefficients was obtained, and reported as an intraclass correlation (ICC) by dividing it by the sum of itself and the residual variance. Simple effects analysis was applied with ANOVA (type III sums of squares) and the Kenward–Roger method for degrees of freedom calculation. Within-subject changes in dependent variables were analyzed using stochastic superiority (a post–pre), which represents the probability that a randomly selected post-intervention score is greater than a randomly selected pre-intervention score [44]. We decided to apply this type of effect size calculation because of the non-normality distribution of our dependent variables and according to Vargha and Delaney [44] the stochastic superiority is a common language effect size that may be directly applied for any discrete or continuous variable that is at least ordinally scaled. The probability values of the stochastic superiority are organized in a qualitative scale from small (0.56–0.64), to medium (0.65–0.71) and large (>0.71) when they tend to 1. The values between 0.44–0.56 are considered negligible. When the values tend to 0 are organized from small (0.43–0.36), to medium (0.37–0.29) and large (<0.29).

The level of significance was set at *p* < 0.05 in all analyses.

## 3. Results

ANOVA omnibus test showed a time effect (*p* < 0.05) in right BKFO (F_1,42_ = 4.60; *p* = 0.038) and both pelvic tilt tests (right F_1,42_ = 22.01, *p* < 0.001; left F_1,42_ = 19.13, *p* < 0.001). There were non-significant interaction effects. The fixed effects estimated parameters for right BKFO showed that both groups had less pressure variation after intervention compared with pre-intervention (β = −1.85 mmHg, 95%CI = [−3.54 to −0.16], t_42_ = −2.14, *p* = 0.038). Furthermore, left pelvic tilt (β = 37.0 s, 95%CI = [20.4 to 53.6], t_42_ = 4.37, *p* < 0.001) improved 8.9 s more than right pelvic tilt (β = 28.1 s, 95%CI = [16.3 to 39.8], t_42_ = 4.69, *p* < 0.001) considering both groups together. Simple effect analysis of group within pre did not show differences between group before the intervention in any of the variable analyzed. However, although there was not interaction in the omnibus ANOVA test, simple effects analysis of time within group revealed that BKFO improved compared with pre only in the EG (right BKFO: β = −3.13 mmHg, 95%CI = [−5.54 to −0.72], t_42_ = −2.62, *p* = 0.012; left BKFO: β = −2.91 mmHg, 95%CI = [−5.27 to −0.56], t_42_ = 2.49, *p* = 0.017).

Changes in the factor group were analyzed throughout the stochastic superiority (a post–pre), which represents the probability that the randomly selected score from the post-intervention would be greater than the intervention. EG reduced their pressure variation with medium and small changes for right and left BKFO, respectively (Table 3). Conversely, CG right and left BKFO did not change. EG improved more than CG for the right (medium vs. small, EG vs. CG) and left (large vs. medium, EG vs. CG) pelvic tilt test.

Table 3 shows the estimated marginal means and their 95% CI of both groups’ dependent variables. Significance of the simple effect of the time factor and the stochastic superiority of post- vs. pre-intervention (Apost-pre) have been calculated. The variance of the random coefficients is represented as an intraclass correlation (ICC, variance of random component divided by the sum of itself and the residual variance).

Here, below, Figure 3 shows individual CS test changes pre- and post-intervention. The analysis of the random component shows a greater variance between subjects concerning the residual variance in the case of the pelvic tilt test right (ICC = 0.74, *p* < 0.001) and it was not the case for the left side and the rest of the tests.

The average sRPE values were calculated to ensure that no load training differences existed among groups. The lowest and the highest values range of all sessions and all gymnasts were 1050–1358 a.u. for the EG, and 958–1335 a.u. for the CG. There were no significant differences between groups.

## 4. Discussion

The main purpose of this study was to analyze the effectiveness of an eight-week integrated functional CST program on the CS of young rhythmic gymnasts. EG improved in four tests: BKFO right and left and pelvic tilt right and left. However, CG only improved in both sides in the pelvic tilt test, although differences between the groups were not significant, and there was a lack of interaction, which means that there was no effect of time depending on the group the gymnasts were allocated. To our knowledge, this is the first study measuring CS using PBU tests and the pelvic tilt test in young rhythmic gymnasts. The main reason to do the test that we chose were that, in young RG it is much more important to focus the CST on the lumbopelvic control instead of the core strength. The PBU is an unexpensive device that can help practitioners to have an objective data about the ability of the RG to keep their lumbopelvic area stable while they are moving their lower limbs, as they do on their sport. Additionally, we add the Pelvic Tilt test to check the endurance of the muscles who are responsible to maintaining the lumbopelvic area aligned. All these tests gave us valid information about the LPMC status in the RG and also guarantied the ecological validity of the study.

These results suggest that adding a functional CST to regular training showed a trend in improving the performance of CS-related variables, which could help improve RG-specific performance. Our findings agree with the CST study implemented on rhythmic gymnasts [25]. The authors concluded that CST is an effective plan for improving the physical and technical characteristics of young female Malaysian rhythmic gymnasts. Similarly, another recent study reported core strength improvements in young rhythmic gymnasts after a twelve-week traditional CST [26]. Nonetheless, these authors did not carry out an integrated functional CST for RG. The authors of an artistic gymnasts’ study concluded that core endurance was improved after an 8-week traditional CST [45]. However, this study likewise implemented traditional CST instead of functional CST training. The study implemented in dancers [46] reported that an intensive nine-week training CS program improved dance performance, balance, and core muscle. It should be noted that this study designed a functional CST for dancers and measured core and performance parameters which adds interest to the results. Nevertheless, RG has specific elements and rules; hence, these findings should be tested and adapted to rhythmic gymnasts.

The results indicated significant individual variability in the BKFO right and left tests for the EG compared with the CG after the CST intervention. The EG showed a medium improvement for BKFO right test and a small improvement for the BKFO left, whereas CG showed negligible results for both right and left BKFO tests. The BKFO is a reliable test to assess LPMC by performing an external rotation of the hip [47]. This test indicates the ability to lift and rotate the lower limb in the horizontal and frontal plane and stabilize the lumbopelvic posture [39]. The EG achieved better results in these two tests; therefore, we could assume that the functional CST helps improve motor control over the gymnasts’ rotational and abduction lower limb movements. These improvements may be transferred to better execution of the rond de jambe movements (a round of the leg), fouettés (a whipping movement), side battements (a large rapid leg kick to the side), and RG balances that include these movements, widespread techniques used in RG that contain rotational and abduction lower limb movements [5]. To our knowledge, this study was the first to try to assess BKFO with the PBU device in young rhythmic gymnasts; thus, the comparison with other cross-sectional or interventional studies is not possible.

EG improved more in the pelvic tilt test compared with other tests after the eight-week CST intervention. The EG showed large improvement for left pelvic tilt and medium improvement for right pelvic tilt, whereas CG reached medium and small improvement, respectively. The Pelvic Tilt is a CS test to measure endurance of the core [40]. Core muscles need endurance to hold postures and to keep the lumbopelvic area stable, consequently, exercises that compromise core endurance should be encouraged [37]. This suggests that the functional CST program helps improve core endurance, especially on the left leg but also indicates that the traditional RG training enhances gymnasts’ core endurance. In the traditional training, gymnasts perform skill repetitions with their preferred leg, usually the right leg, whereas in the functional CST, gymnasts use both legs equally. This might be an explanation for higher improvements on the Pelvic Tilt left tests in the EG compared with the CG. According to our results, in the artistic gymnasts’ study, EG improved in the supine plank test (similar to the Pelvic Tilt test position). In contrast, the CST was traditional and not functional, and the CG did not improve in the core endurance tests [45]. It should be noticed that core endurance is an integral component of the complex factors that comprise balance performance [8]. The study conducted in dancers indicated that improvements in abdominal muscle strength through CST, lead to improved static and dynamic balance and the spinning ability for pirouettes [46]. This suggests that functional CST could provide the gymnasts with higher levels of balance, one of the three fundamental groups of body elements included in RG competition routines [5]. To our knowledge, this study was the first to assess the effects of a functional CST in the pelvic tilt test in young rhythmic gymnasts; thus, the comparison with other cross-sectional or interventional studies is not possible.

In contrast, EG and CG did not show significant differences or improvements regarding the ASLR right and left tests after the eight-week integrated CST program. The ASLR is a valid test to evaluate LPMC [36] measuring hip flexion control in sagittal plane and lumbar motor control [39]. This finding suggests that neither traditional RG training nor CST challenges hip flexion control in sagittal plane movements and lumbar motor control. In addition, the CG showed inferior results in the ASLR left test after the eight weeks.

It is important to note that there is great individual variability in the results of the two groups (Figure 2), especially in the BKFO and the ASLR tests. The pelvic tilt test is much more consistent, and it presents more responders than the other two tests. the ICC of these tests can be observed also in Table 3. This suggests that the pelvic tilt test is more objective to measure CS in RG than the BKFO and ASLR tests. Consequently, these results could also mean that the PBU tests present higher variability due to their characteristics. Even though recent studies have reported the validity of the PBU tests [36,37,39,48,49], it is also discussed that no CS test serves as a gold standard [50]. In our case, the BKFO and ASLR tests might be less consistent than the pelvic tilt to better capture the effects and enhancements of both CS and traditional RG training.

Our functional CST is innovative as it is linked to RG motor patterns. In our study gymnasts trained core exercises with a demand for balance, postural control, and SSC explosive strength. The challenges imposed on CS are close to the demands of the RG skills. This statement is supported by Lederman [18] who mentioned that respecting the sports functionality in CST is necessary to improve sports performance. The most specific is the training, more transfer to the sport will be achieved. It is necessary to approach the training sport’s reality and respect, in our case, the functional necessities of RG. Consequently, this program is considered more functional than a traditional CST because it is a holistic design that includes important RG performance determinants. Our results show a bigger effect size in the EG in the BKFO and especially in the Pelvic Tilt test, this suggests that the CST tends to improve CS measures; however, more specific RG tests should be conducted to assess that this functional CST improves RG performance.

As we mentioned, this specific training integrated core control demands while performing core strength actions, challenging balance and postural control in specific RG elements. Previous research supported integrated CST showing that neuromuscular control can be enhanced by joint stability exercises, balance training, perturbation training and plyometric, or jump exercises [51]. Our training includes this type of exercises combined and the results suggest that an integrated CST improves core motor control and edurance.

This training can be also considered physical preparation training given that it contains trunk isometric strength and lower limbs explosive strength exercises. It has been mentioned that a proper level of physical performance is a prerequisite for excellent technical performance in RG [4]. In particular, the specific physical preparation of a gymnast develops both her physical fitness and the ability to master RG exercises [52]. Consequently, achieving better core endurance in the EG suggests that this CST helps improve RG’s physical fitness. Furthermore, our CST may help prevent rhythmic gymnasts from injury since CST has been associated with preventing and reducing the risk of injury [12,14,53,54]. A deficient neuromuscular core control predisposes athletes to low back pain and lower extremities injuries [29] and proximal stabilization decreases the risk of lower limb injuries in athletes [9]. Moreover, a study of an eight-week specific core training program implemented in gymnasts, concluded that CS may help prevent and reduce low back pain in this population and recommend CST to be part of the training sessions [55]. Athletes possessing higher levels of CS have less injury risk, therefore, they do not disrupt their training program [50].

Studies conducted with children should be aware of the importance of biological maturity and its relationship to growth and physical performance. Furthermore, to study the effects of sports training we should know how much of the improvement is due to growth and maturation-related changes and how much it reflects training adaptations [56]. In our study gymnasts’ maturity and growth were assessed showing a premenarchal status and a negative PHV age of most gymnasts. The random group division (EG and CG) showed no PHV age significant differences between groups, therefore no PHV age group division was needed. According to Georgopoulos et al. [57], any assessment of sexual maturation must consider the biological indicators of bone age and PHV.

Certain issues and limitations regarding the design are found. Adding more CS exercises on unstable surfaces could further improve CS, since it has been shown that unstable surfaces lead to improved CS and balance by increasing the demands on trunk muscles [29]. Due to the lack of possibility to acquire unstable material for all gymnasts in the EG, unstable surfaces were introduced just in three exercises of the first block of the CST. Given that the CST includes twenty-five exercises, the use of unstable surfaces in this study may not be sufficient to challenge CS. Further research should be conducted to test the effect of unstable surfaces on rhythmic gymnasts’ CS performance while performing a functional CST. Furthermore, to mark the exact RPEs may be more difficult for young children. Regarding the training length, previous CS programs carried out with athletes reported results after interventions of four to twelve weeks [23,28,58]. We opted for an eight-week training as it is in the average CST length, furthermore, it respected the programming logistics of the RG competitive club. Perhaps longer interventions acquire greater adaptations since motor control may need longer training periods [39]. We encourage future works to study the effect of longer CST interventions on CS in rhythmic gymnasts.

To our knowledge, there are only two studies measuring CS after a CST program among rhythmic gymnasts [25,26]. Most available research is focused on conducting CST for rehabilitation, preventing, and reducing the risk of injury in sports [37]. It is suggested that CST might be beneficial for RG performance; however, much more research is needed to develop valid and reliable CS tests that are easy to use by researchers to assess the effects of CST in gymnastics, especially its effects on performance; to evaluate the efficacy of various CST exercises, and to determine the long-term effects of CST.

Moreover, this study aimed to carry out a second intervention program permitting the CG to perform the functional CST. The cross-over design could not be finished due to the COVID-19 pandemic.

## 5. Conclusions

Traditional RG training seems to enhance core endurance; however, core motor control exercises that include flexion and extension exercises in the sagittal plane and core motor control exercises containing hips rotation and abduction movements in the frontal and horizontal planes are lacking in regular training programs for gymnasts. The integrated functional CST program presented in this work led to a greater improvement of rhythmic gymnasts’ core endurance and a slight improvement in overall CS performance. Hence, adding a functional CST to regular training could help to improve the performance of CS-related variables. Nonetheless, future investigations should verify the effectiveness of the functional CST in RG performance, evaluating the effects on specific RG skills.

## Figures and Tables

**Figure 1 ijerph-19-03509-f001:**
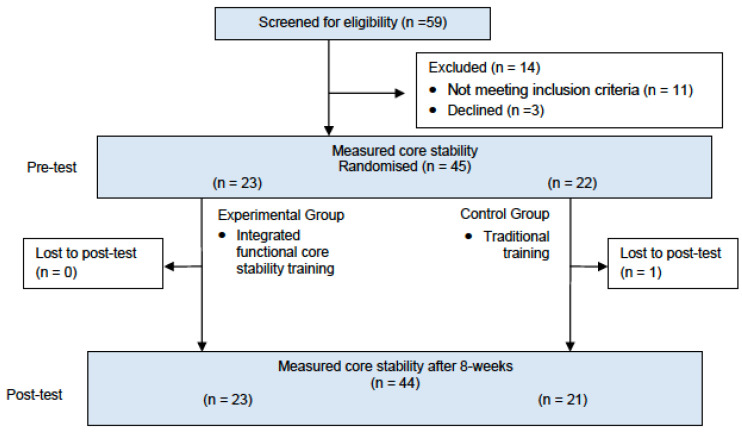
Flowchart.

**Figure 2 ijerph-19-03509-f002:**
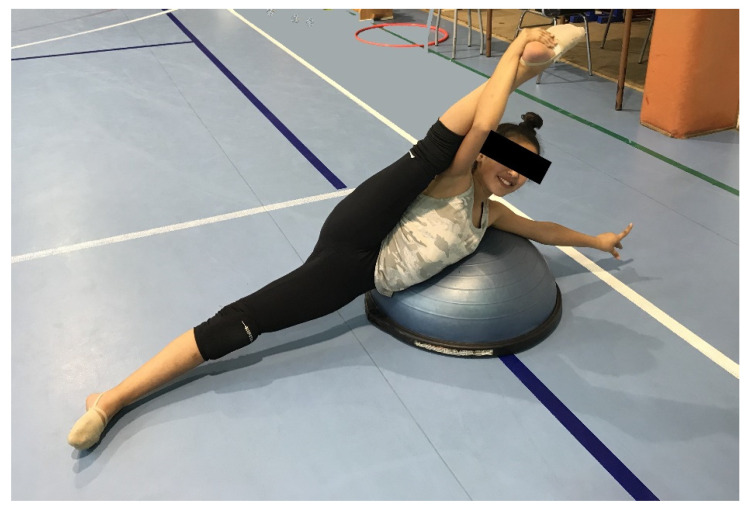
Example of one RG functional CS exercise.

**Figure 3 ijerph-19-03509-f003:**
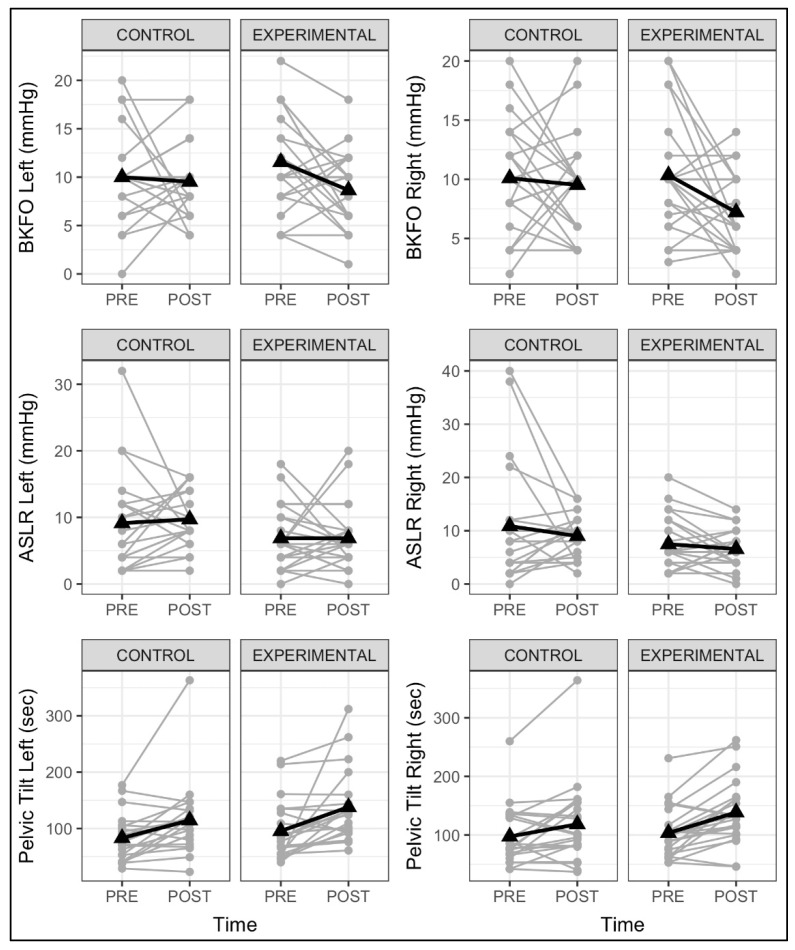
Individual variability results in all CS tests.

**Table 1 ijerph-19-03509-t001:** Subjects’ Characteristics.

Variables	EG (n = 23)	CG (n = 21)	*t*-Test (*p*-Value)
Age (years)	10.52 ± 1.90	10.43 ± 1.78	0.874
Peak height velocity (PHV) (years)	12.25 ± 0.55	12.23 ± 0.67	0.917
Years from PHV	−1.21 ± 1.41	−1.10 ± 1.38	0.798
Height (m)	1.44 ± 0.10	1.44 ± 0.11	0.801
Weight (kg)	37.82 ± 9.83	38.2 ± 8.03	0.892
Body mass index (kg/m²)	18.08 ± 2.56	18.06 ± 1.56	0.982
Gravity center height (m)	0.87 ± 0.07	0.87 ± 0.07	0.922

**Table 2 ijerph-19-03509-t002:** Overview of the 8-week functional CST program.

Exercises	W1		W2		W3		W4		W5		W6		W7		W8	
	Sets	Rep	Sets	Rep	Sets	Rep	Sets	Rep	Sets	Rep	Sets	Rep	Sets	Rep	Sets	Rep
Block 1																
Lateral hip bridge over bossu	2	6	2	6	3	6	3	6	3	6	3	6	4	6	4	6
Prone plank over ball	3	6	3	6	3	6	3	6	3	6	3	6	4	6	4	6
Plyometric jumps + RG balances	2	6	2	6	2	6	2	6	2	6	2	6	2	6	3	6
Plyometric double jumps + RG balances	2	6	2	6	2	6	2	6	2	6	2	6	2	6	3	6
Plyometric RG jumps with cones	2	8	2	8	4	8	4	8	4	8	6	8	6	8	6	8
Plyometric RG jumps + hurdles	4	8	4	8	4	8	4	8	4	8	6	8	6	8	6	8
RG balances over balance disc	2	3 × 5″	2	3 × 5″	2	3 × 5″	2	3 × 5″	2	3 × 5″	2	3 × 10″	2	3 × 10″	2	3 × 10″
Block 2																
DJ + passe balance	1	2	1	2	1	2	1	2	1	2	1	2	1	2	1	2
DJ + arabesque balance	1	2	1	2	1	2	1	2	1	2	1	2	1	2	1	2
JDJ + passe balance	1	2	1	2	1	2	1	2	1	2	1	2	1	2	1	2
JDJ + arabesque balance	1	2	1	2	1	2	1	2	1	2	1	2	1	2	1	2
JDJ + side balance	0	0	0	0	1	2	1	2	1	2	1	2	1	2	1	2
DJ + stag jump	1	2	1	2	1	2	1	2	1	2	1	4	1	4	1	4
DJ + scissors jump	1	2	1	2	1	2	1	2	1	2	1	4	1	4	1	4
DJ + split leap	1	2	1	2	1	2	1	2	1	2	1	4	1	4	1	4
Block 3																
Lateral plank arm straight	2	6	2	6	2	6	2	6	2	6	4	6	4	6	4	6
Lateral plank with elbows	2	6	2	6	2	6	2	6	2	6	4	6	4	6	4	6
Hip bridge with passé balance	2	4 + 8″	2	4 + 8″	2	4 + 16″	2	4 + 16″	2	4 + 16″	2	4 + 16″	2	4 + 16″	2	4 + 16″
Hip bridge with side balance	2	4 + 8″	2	4 + 8″	2	4 + 16″	2	4 + 16″	2	4 + 16″	2	4 + 16″	2	4 + 16″	2	4 + 16″
Hip bridge with passé balance + bounces	2	4	2	4	2	8	2	8	2	8	2	8	2	10	2	10
Hip bridge with side balance + bounces	2	4	2	4	2	8	2	8	2	8	2	8	2	10	2	10
Passé balance + jumps	2	4	2	4	2	8	2	8	2	8	2	8	2	10	2	10
Arabesque balance + jumps	2	4	2	4	2	8	2	8	2	8	2	8	2	10	2	10
Side balance + jumps	2	4	2	4	2	8	2	8	2	8	2	8	2	10	2	10

DJ = drop jump, JDJ = jump-drop-jump, W = week. Block 1. circuit = mixed CS, balance, and plyometric exercises. All exercises were performed with the right and left sides. All planks and lateral hip bridges were executed with passe, side leg, and arabesque positions and were maintained for 2 s in each position. All DJ and JDJ were performed with a 30 cm bench.

**Table 3 ijerph-19-03509-t003:** CS test results.

	EG		Apost-Pre	*p*-Value	CG		Apost-Pre	*p*-Value	*p* Value Time × Group
	Pre-	Post-			Pre-	Post-			ICC Random Components
BKFO Right	10.4 [8.5;12.1]	7.2 [5.4;9.0]	0.69 medium	0.01 *	10.1 [8.2;11.9]	9.5 [7.6;11.4]	0.53 negligible	0.64	0.14
									0.13
BKFO Left	11.6 [9.7;13.4]	8.6 [6.7;10.5]	0.66 small	0.01 *	10.0 [8.0;11.9]	9.5 [7.5;11.5]	0.55 negligible	0.69	0.15
									0.22
ASLR Right	7.5 [4.7;10.2]	6.6 [3.8;9.3]	0.52 negligible	0.59	10.9 [7.9;13.7]	9.0 [6.1;11.8]	0.44 negligible	0.30	0.71
									0.24
ASLR Left	6.9 [4.5;9.1]	6.9 [4.6;9.4]	0.49 negligible	0.47	9.1 [6.7;11.5]	9.7 [7.3;12.0]	0.40 small	0.33	0.78
									0.22
Pelvic Tilt Right	103.4 [80.2;126.5]	138.6 [115.4;161.7]	0.29 medium	0.001 **	97.3 [73.1;121.5]	118.3 [94.0;142.5]	0.38 small	0.02 *	0.24
									0.74
Pelvic Tilt Left	95.5 [72.3;118.6]	137.9 [114.7;161.1]	0.25 large	0.001 **	83.0 [58.7;107.2]	114.6 [90.3;138.8]	0.32 medium	0.01 *	0.52
									0.49

Estimated marginal means and 95%CI for bent knee fall out (BKFO) test, active straight leg raise (ASLR) test, and pelvic tilt test. All tests were evaluated right and left sides. A post–pre stochastic superiority (probability and qualitative inference) of post-intervention versus pre-intervention assessment. Significance: * *p* < 0.05, ** *p* < 0.001.

## Data Availability

All data files are available from the figshare database: https://doi.org/10.6084/m9.figshare.16810885.v1 (accessed on 29 January 2022).

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
