# Peer review of "The Effect of Eight-Week Functional Core Training on Core Stability in Young Rhythmic Gymnasts: A Randomized Clinical Trial"

_ijerph, 2022, doi:10.3390/ijerph19063509_

Round 1
Reviewer 1 Report
The effect of eight-week functional core training on core stabil- 2
ity in young rhythmic gymnasts: A randomized clinical trial,.
After the substantial changes made by the authors, which greatly strengthened the excellent manuscript carried out. In this way, I congratulations to authors for the excellent manuscript, and this reviewer who writes to you only has to ACCEPT the work for publication in the IJERPH.
Author Response
We greatly appreciate your acceptance of this manuscript as well as your time invested in analyzing our study.
Reviewer 2 Report
The authors' corrections have been confirmed. I consider the content has been greatly improved.
Minor comments are provided below.
Line 70-75.
This sentence is an introduction to previous research on CST, but I do not understand what is being done and what the problem is. A brief review of the previous studies is needed.
L. 88-92.
For the sample size, did you calculate it based on the fact that you will analyze it by ANOVA?
Also, what was the effect size of 0.4 chosen based on?
Author Response
Please note that:
[R2] = comments from Reviewer #2.
[A] = answers from the authors.
{…} = text modified in the revised manuscript.
[R2] =
The authors' corrections have been confirmed. I consider the content has been greatly improved.
Minor comments are provided below.
Line 70-75.
This sentence is an introduction to previous research on CST, but I do not understand what is being done and what the problem is. A brief review of the previous studies is needed.
[A] =
We have amended as follows: {...Nevertheless, it is accepted that there is still a lack of more functional CST programs and assessments with sufficiently sensitive measurement protocols [31].}
- 88-92.
For the sample size, did you calculate it based on the fact that you will analyze it by ANOVA?
Also, what was the effect size of 0.4 chosen based on?
[A] =
The sample size was calculated to have a prior reference of the number of participants necessary to carry out a consistent study, without taking into account the subsequent statistical analysis.
The effect size was set at 0.4 since it is the point at which it begins to go from small to moderate, and we thought that it was the minimum value to consider in this study. Similar previous investigations obtained effect sizes with a range of 0.5-0.7 (Esteban-Garcia, 2021) and 0.5-0.6 (Bon Hoi, 2019)
We greatly appreciate all your reviews and suggestions, as well as your time invested in analyzing our study.
Reviewer 3 Report
Thank you for addressing my comments. The manuscript has been improved, however, it is still lacking in the way it is structured and written.
- There was no significant difference between groups (line 320-321), but the authors concluded that “ adding a functional CST to regular training improved CS performance leading to increased specific RG performance”. Why? As a matter of fact, the Discussion should offer an interpretation based on the findings.
- I do not understand why the authors used ICC, which is used to assess the reliability of measurements in the same group. Considering the research purpose, this additional arrangement detracts from the clarity of the message conveyed.
- Introduction, it is nice to notice that this section has a big improvement. Yet, I think it is needed to point out the rationale of the training duration of 8-week functional core training.
- Uncommon expression of “Two PBU tests were applied to evaluate LPMC:” (line199)
- “will be greater than the intervention” or “would be greater than the intervention” (line 290)?
- The statement, ‘The PBU is an inexpensive device that can help practitioners to have objective data about the ability of the RG to keep their lumbopelvic area stable while they are moving their lower 327 limbs, as they do on their sport. Additionally, we add the Pelvic Tilt test to check the endurance of the muscles that are responsible for maintaining the lumbopelvic area aligned. All these tests gave us valid information about the LPMC status in the RG and also guaranteed the ecological validity of the study (line 325-330), which are not relevant to the research hypothesis of this study. It might be placed in the Methods.
Author Response
Please note that:
[R3] = comments from Reviewer #3.
[A] = answers from the authors.
{…} = text modified in the revised manuscript.
[R3] =
Thank you for addressing my comments. The manuscript has been improved, however, it is still lacking in the way it is structured and written.
There was no significant difference between groups (lines 320-321), but the authors concluded that “ adding a functional CST to regular training improved CS performance leading to increased specific RG performance”. Why? As a matter of fact, the Discussion should offer an interpretation based on the findings.
[A] =
We have changed the sentence as follows to be more consistent with the results {...Adding a functional CST to regular training showed a trend in improving the performance of CS-related variables, which could help improve RG-specific performance} In the abstract conclusion, in the discussion lines 348-359, and the conclusions section lines 501-502.
[R3] =
I do not understand why the authors used ICC, which is used to assess the reliability of measurements in the same group. Considering the research purpose, this additional arrangement detracts from the clarity of the message conveyed.
[A] = The reviewer is correct, the ICC is traditionally used to assess the reliability of two measures. In our case, the ICC was not used traditionally, in our study the ICC was used to report the variance of the random coefficients.
In this type of analysis, the ICC is a way of reporting the variance and is obtained by dividing it by the sum of itself and the residual variance. A higher ICC indicates a greater variance, this can be very useful in the sense that the intervention affects each subject but takes into account the initial state of it. The use of the ICC in this sense is explained in the statistical analysis section.
[R3] =
Introduction, it is nice to notice that this section has a big improvement. Yet, I think it is needed to point out the rationale of the training duration of 8-week functional core training.
[A] =
We add a sentence in the introduction as follows: {...Several studies evaluated the effect of CST on sports performance, most of them are purposes from 4 to 12-week length, 8-week programs are average length [20–27].}
We reviewed the cited literature, adding studies conducting CST programs ranging from 4 to 12 weeks.
[R3] =
Uncommon expression of “Two PBU tests were applied to evaluate LPMC:” (line199).
[A] =
We have amended as follows: {...There were performed two PBU tests to assess LPMC:...}
[R3] =
“will be greater than the intervention” or “would be greater than the intervention” (line 290)?
[A] = Thank you, amended.
[R3] =
The statement, ‘The PBU is an inexpensive device that can help practitioners to have objective data about the ability of the RG to keep their lumbopelvic area stable while they are moving their lower 327 limbs, as they do on their sport. Additionally, we add the Pelvic Tilt test to check the endurance of the muscles that are responsible for maintaining the lumbopelvic area aligned. All these tests gave us valid information about the LPMC status in the RG and also guaranteed the ecological validity of the study (lines 325-330), which are not relevant to the research hypothesis of this study. It might be placed in the Methods.
[A] = Thank you, amended in lines 203-206 and lines 228-230 in the methods section.
We greatly appreciate all your reviews and suggestions, as well as your time invested in analyzing our study.
Reviewer 4 Report
Thanks for sending the revision. Although the authors attempt to revise the manuscript to meet the standards of accepting form, the methodological considerations I raised still remain concern. The authors failed to clarify the technical issues for the limitations of the study. Therefore, I do not recommend this study for publication in IJERPH.
Author Response
Please note that:
[R4]= comments from Reviewer #4
[A] = answers from the authors.
[R4] =
Thanks for sending the revision. Although the authors attempt to revise the manuscript to meet the standards of accepting form, the methodological considerations I raised still remain concern. The authors failed to clarify the technical issues for the limitations of the study. Therefore, I do not recommend this study for publication in IJERPH.
[A] = We regret your opinion, as far as we have been able to justify and clarify the methodological issues that you demanded in the previous review. However, to the best of our knowledge, we do not believe that the methodology used is a limitation but rather a different alternative for this type of study. We appreciate your time invested in analyzing our study.
Reviewer 5 Report
The authors have made substantial improvements to the manuscript and addressed most of the issues. I believe although there are limitations to the study and design, that if they're presented to the reader, as is the case at the end of the discussion, it is up to the reader to take these into consideration.
Author Response

(The authors gave the same response as above.)

Round 2
Reviewer 3 Report
I have no more concerns.
Reviewer 4 Report
N/A
This manuscript is a resubmission of an earlier submission. The following is a list of the peer review reports and author responses from that submission.
Round 1
Reviewer 1 Report
THE EFFECT OF EIGHT-WEEK FUNCTIONAL CORE TRAINING ON CORE STABILITY IN YOUNG RHYTHMIC GYMNASTS: A RANDOMIZED CLINICAL TRIAL
General Commentary
This article presents a very interesting and pertinent question of research of the investigate effect of a functional core stability training on young rhythmics gymnasts’ core stability performance.
However, some questions need to be clarified in order to better understand and apply the results found.
MAJOR CONSIDERATION
ABSTRACT
I understand what "bent knee fall out" means. However, the authors describe in the results mean (e.g., 0.69) and small (e.g., 0.66) differences, what do such values mean, is it a scale? more or less in relation to the control group? what is the level of significance? Anyway, I suggest rewriting all ABSTRACT result.
METHODS
Table 1 and Figure 1
Table 1 and Figure 1 were superimposed, I can't identify anything about the two. The following figure can see something.
Participants
As I cannot visualize figure 1, it was not clear why the experimental group had 23 subjects and the control with 21? In the case of randomization, shouldn't they have N equal? Do you hear sample loss?
Measurements
Line 220
I didn't find in the text what the PBU means, please correct it if necessary.
Tests
Lines 220-269
Please describe in each of the tests how the analyzes were carried out, and if necessary, how they were classified, obviously citing references about it.
Statistical
As the authors have two groups, being evaluated in two moments, the actors should use a statistical test with two factors (e.g., ANOVA two way; MANOVA Two Way...). Please justify the statistic used, or change the statistic tests used.
Lines 278-286
The authors rank the probability of the stochastic superiority. However, where did this classification come from? for which tests is it validated?
RESULTS
When observing the results presented in table 1, the classification described above was not clear. Example: Final line (0.25 = Large); Line 3 (0.52 = negligible) and Line 1 (0.69 medium). Shouldn't this classification take into account the value of P? The value of P indicates whether it is significant or not, correct? But doesn't the classification adopted depend on the value of P? or depends?
In relation to the last column of the table, what are the presented values (P)? or r of the ICC? please correct the table.
Table 1.
The authors present the significant differences or not between the moments of each group. But at no time were the comparisons between the Groups presented in the Pre and Post moments.
The authors present the intraclass correlation coefficient (ICC) values (r) in the results. However, in none of the methods was such ICC classified. Furthermore, I emphasize that this test is not a comparison, but an association between two measures of the same variable, for example. That is, authors cannot compare groups or moments with the ICC test.
Again, I emphasize that I did not understand the statistical tests used to compare groups and moments. So, do I ask the authors for an adequate explanation of such tests used?
As far as my review is concerned, the tests used are inadequate for such comparisons of groups and moments, that is, the authors must redo all the analysis and statistics of the data and then re-present the results from the comparison tests, as mentioned above, otherwise the objectives and results and the rest of the manuscript must be rewritten based on reliability tests of measures, such as the ICC, and not on comparisons between groups and moments.
DISCUSSION
CONCLUSION
The discussion and conclusion of the manuscript depend on what was mentioned earlier in the comments. Please calmly review everything that has been done and modify accordingly if necessary.
Reviewer 2 Report
I think this is an important finding regarding the usefulness of CST, but I see many things that need to be corrected in the structure of the paper, and some things that are not related to the research content are mentioned.
Abstract
- 16–17
Does CST refer to Functional CST?
I think there are several studies on the effect of CST on performance. If you want to differentiate between Functional CST and Traditional CST, you need to add an explanation of that.
- 31–33.
I think probably misstated. Is not the following correct?
The pelvic Tilt test right and left tests showed medium (Apost-pre 0.29) and large (Apost-pre 0.25) effects in the EG and small to medium values (Apost-pre right 0.38 and left 0.32) in the CG.
Introduction
The whole thing is long and gives the impression that the story is poorly structured. With the current content, it is not clear why readers need to look at the effects of functional CST on rhythmic gymnasts.
Since this study is examining the effects of functional CST, it is necessary to add an explanation of what functional CST is and what its advantages are compared to traditional CST.
- 105–109.
This sentence is not necessary for the Introduction. Please move to the method.
Method
Since this is an RCT study, you need to add a description according to CONSORT. Such as blind, sample size, trial design. You state in the statics section that you used GPOWER to calculate the sample size, but I could not find anything about sample size results.
Figure 1 overlaps Table 1 and I cannot see Table 1.
- 151
Please add the approval number of the ethics committee.
- 279
Please change “95% confidence intervals to 95% CIs”.
Results
- 301
Please change “than the intervention” to “than the pre-intervention”.
Discussion
Throughout the discussion, it is too long and includes content that is not relevant to this study. The description needs to be revised to focus on what is relevant to this study.
- 433–439
There is a sudden mention of the multidisciplinary approach, but how does it relate to this study? I do not understand the relationship between this study and the multidisciplinary approach.
- 447–461
I understand that CST is beneficial in reducing injuries, but what is the relevance to this study? The same goes for Extra weight.
- 494–508.
This is a repetition of the Introduction. What are the limitations of this study? Describe what the problem is and what methods would be desirable to solve it.
Reviewer 3 Report
The purpose of this study was to examine the effect of functional core stability training (CST) on core stability (CS) performance in young rhythmics gymnasts. The findings found that a functional CST improved the CS in young rhythmic gymnasts. Coaches working with rhythmic gymnasts should consider adding a functional CST to regular training to improve CS performance. I think the manuscript quality has a big room to improve the defects such as tediously long with irrelevant information, not well-developed rationale, unfavorably documented method, and unsatisfied data interpretation.
Specially concerns
Abstract
-No need to list the abbreviations of all term in this section (i.e. PHV). Also, I do think the information of PHV can be removed.
-Unclear descriptions of the results.
Introduction
-The authors need to state the rationale of the present study systemically. The present manuscript does not provide a clear summary of the current studies and what key unanswered questions this study is trying to address. It is unreasonable to address the debate on the definition of the core (pare 2), analysis of posture biomechanics (para 3), and so on such irrelevant information (vague purpose of para 3).
-Some important information is missed. Why was the 8-week functional core training carried out?
- Any evidence for the hypothesis?
Material and Methods
-Repeated statement of “Helsinki Declaration” (Line 147-152).
-Why is PHV used to the maturational status? Tanner’s criteria seem more common.
- Repeated phrase of “A manual chronometer (Namaste© model 898, Spain)” (Line 235, 259).
- Please provide the reliability of the measures.
-Statistical analyses, any references for the definition of “the probability values of 282 the stochastic superiority” and the estimation of sample size? What is the result of the sample size? How to calculate ICC, which was used later?
-Table 2 & 3 provide limited information. It is better to place these two tables as supplemental materials.
Result
-Unclear statement of the results. Please provide the values of F, ƞ2 and p values of the two-way mixed ANOVA and then report whether there are a group x time interaction effect, main time and group effects. Indeed I’d like to suggest the later (one-way ANCOVA), whereas the baseline values of outcomes are taken in as a covariate in the analyses of within-group changes (Dimitrov & Rumrill, Work 2003; Vickers & Altman, BMJ 2001), especially the pre-measures were not the same in the present study (Table 4).
Discussion
-Overall, this section is superficial and loosely organized. Please revise this session and interpret the results concisely and comprehensively.
- The conclusion section does not seem to follow the results of this study well.
Reviewer 4 Report
This study investigated effects of 8-weeks core training intervention on children rhythmic gymnasts. The research design is based on parallel randomized trial. The results revealed significant improvements of performance during bent knee fall out and pelvic tilt tests. However, the methods to assess the core muscles strength and its association to functional performance in rhythmic gymnast are bias. Moreover, the training programme facilitated in the training group is not mainly focused on core muscle strength but also power performance. Based on the considerations below, I cannot recommend the current form of this paper for publication.
Major comments
The major bias in this study is that the authors used the pressure biofeedback unit to assess the lumbopelvic stability. This assessment is mainly used to assess the lumbar/pelvic health in relation to pathological conditions (e.g. low back pain). The exercise training programme performed in this study consisted of core muscle strength and polymetric exercises. Such assessment cannot reflect the chronic training adaptation in this study.
The second major concern in this study is related to similar training regimen engaged in training and control groups during the study period. In Table 3, it seems both groups undertook same sports training with different exercise activities during 30 min studying break. It is unclear what difference of exercise loads between the training intervention and rhythmic gymnast warm up activities, particular the authors recorded sRPE (no information shown in this paper).
The third major concern is the statistical comparison by using stochastic superiority. As showed between line 282-286, It is unclear how the authors defined the levels of probability. The authors are suggested to examine the magnitude of effect size or small worthwhile change.
Minor comments
- The CONSORT statement should be presented as a randomized control study.
- Sample size estimation should be descripted in Participants section.
- The Figure 1 and Table 1 are overlapped.
- The data presented in Table 4 and Figure 2 is duplicated.
Reviewer 5 Report
The authors have clearly put a lot of time and effort into a very applied piece of research in gymnastics and should be commended for doing so. Whilst I enjoyed the manuscript there are a number of issues I feel need addressed and some improvements can be made. Please see below
Intro:
L59-60 change the sentence with ‘signaled’ to “The importance of the core in providing trunk strength and balance has been outlined previously [7]” for readability.
L62 change ‘exposed’ to ‘combined’ or ‘taken together’
L63-64 change ‘highly demanded’ to ‘required’
L65 please clarify what ‘high-level activities’ are as this is a vague term. Change ‘be enabling’ to ‘enable’ or ‘facilitate’
L69-70 the core redistributes force, forces can’t be ‘absorbed’ by the body
I believe the 3rd paragraph (L64-89 can be made more succinct
L94-96 Terms ‘core endurance’ and ‘core performance’ are very vague and ambiguous. How are these defined/ measured?
I think this paragraph (L105-109) needs to spend more time outlining the measurement aspect of ‘core’ function. This is critical to the validity of the study.
Methods
Figure 1 appears over the text of Table 1. Please amend.
Study design – why did the experimental group do explosive training in addition to CST whereas controls did traditional training? There is now a confounding factor because the CST is not the only program variable that is different between groups.
L148 Declaration of Helsinki
The authors should be commended on the detailed overview of the entire training program
Results
Figure 2 appears to be somewhat replicative of data in table 4. Please remove figure 2 data from table 4. Figure 2 also requires significance symbols etc to be added to the panels. For all tables and figures please ensure titles are full and accurate descriptions.
Discussion
As in the methods its stated that pelvic tilt test assesses both ‘core strength and endurance’. It cant be both im afraid so which is it? Yes endurance will be affected by strength but the authors have no way of differentiating from this test that both of those factors have been altered. I think authors should refer to this as endurance only.
L357 change ‘tight’ to ‘right’
The discussion is reasonable however the authors have not outlined in the limitations the fact that the training programs were different in two ways – CST and explosive training vs. traditional. The authors need to explore the fact that they cannot discount some potential CS improvements could have come from the explosive training.